# Terrestrial and Airborne Structure from Motion Photogrammetry Applied for Change Detection within a Sinkhole in Thuringia, Germany

Helene Petschko *, Markus Zehner, Patrick Fischer and Jason Goetz

Department of Geography, Friedrich Schiller University Jena, Loebdergraben 32, 07743 Jena, Germany; markus.zehner@uni-jena.de (M.Z.); patrick.fischer@uni-jena.de (P.F.); jason.goetz@uni-jena.de (J.G.)
* Correspondence: helene.petschko@uni-jena.de

**Abstract:** Detection of geomorphological changes based on structure from motion (SfM) photogrammetry is highly dependent on the quality of the 3D reconstruction from high-quality images and the correspondingly derived point precision estimates. For long-term monitoring, it is interesting to know if the resulting 3D point clouds and derived detectable changes over the years are comparable, even though different sensors and data collection methods were applied. Analyzing this, we took images of a sinkhole terrestrially with a Nikon D3000 and aerially with a DJI drone camera in 2017, 2018, and 2019 and computed 3D point clouds and precision maps using Agisoft PhotoScan and the SfM_Georef software. Applying the "multiscale model to model cloud comparison using precision maps" plugin (M3C2-PM) in CloudCompare, we analyzed the differences between the point clouds arising from the different sensors and data collection methods per year. Additionally, we were interested if the patterns of detectable change over the years were comparable between the data collection methods. Overall, we found that the spatial pattern of detectable changes of the sinkhole walls were generally similar between the aerial and terrestrial surveys, which were performed using different sensors and camera locations. Although the terrestrial data collection was easier to perform, there were often challenges due to terrain and vegetation around the sinkhole to safely acquire adequate viewing angles to cover the entire sinkhole, which the aerial survey was able to overcome. The local levels of detection were also considerably lower for point clouds resulting from aerial surveys, likely due to the ability to obtain closer-range imagery within the sinkhole.

**Keywords:** 3D reconstruction; point clouds; precision maps; M3C2; multiscale normals





## 1. Introduction

The possibilities of mapping highly detailed structures or surfaces with fast and cost-efficient 3D reconstruction from optical images is a viable and promising method for small-to medium-scale surveys in geomorphology [1]. This 3D reconstruction is often referred to as structure from motion (SfM) photogrammetry, a term that summarizes a workflow of the application of various algorithms including structure from motion and multiview stereo (MVS) reconstruction. SfM and MVS reconstruction rapidly emerged to a widely applied method for deriving detailed digital 3D terrain representations and for monitoring surface changes in geomorphology and archeology due to its ease of data collection and processing (resulting from today's computing power and software availability) and highly flexible application in various natural settings (e.g., [1–6]). SfM-MVS offers a cost-effective survey method with a comfortable and fast workflow even in hard-to-access study sites [4,7,8]. Originating in the field of photogrammetry and computer vision, structure from motion exploits the stereoscopic view, created by identifying tie points of overlapping images taken from slightly differing positions and viewing angles around an object or structure [9]. Multiview stereo reconstruction then constructs a dense 3D point model using the tie points.

SfM photogrammetry is often employed with and compared to point clouds from LiDAR (light detection and range) from terrestrial (terrestrial laser scanning, TLS) or airborne (airborne laser scanning, ALS) platforms [1,7,10–12]. Given careful ground control placement, calibration, a sufficient number of images, and a suitable surveyed surface (with no or very sparse vegetation), SfM-MVS photogrammetry is able to produce point clouds of resolution equal to or higher than those resulting from TLS or ALS [1,10,13]. The equipment for taking the terrestrial or aerial photos is comparably inexpensive (consumer-grade handheld cameras or even smartphones achieve good results (e.g., [14–16]). However, in densely vegetated areas (dense forest, bushes, or high grass), SfM-MVS can only provide points of the uppermost surface from the sensor's viewpoint, resulting in a surface model rather than a terrain model that is more easily derived from laser scanning data [4,12]. Finally, SfM has been applied at different scales, landscapes, and landforms such as volcanoes and lava movements, large planar regions, glaciers, rock glaciers, badlands, sinkholes, landslides, rivers, burned and aeolian landscapes, as well as at laboratory experiments to detect and analyze surface changes (e.g., [2,17–33]).

Change detection is commonly performed deriving the difference between (1) two point clouds (e.g., cloud to cloud (C2C [34]); multiscale model to model cloud comparison (M3C2 plugin in CloudCompare [35]), (2) a point cloud and a mesh (e.g., cloud to mesh C2M [33]), or (3) two digital elevation models (DEMs in a raster format) computed from point clouds or derived from other remotely sensed data (DEM of difference (DoD) [34]). Nourbakhshbeidokhti et al. [36] provide a comparison of these methods using TLS data to calculate changes in fluvial systems after disturbance and conclude that given complex terrain, M3C2 and DoD are the preferable methods for detecting elevation change and volumetric change, respectively. However, DoD methods might not be able to properly portray areas with overhangs or steep slopes. The M3C2 method has advanced to the standard method for estimating topographic change with assigned levels of detection, as it has been specifically developed for applications in complex terrain, works with both LiDAR and SfM data, provides the possibility to include 3D precision estimates for each point of the point cloud, and is implemented within the software CloudCompare [35,37].

Three-dimensional point clouds resulting from SfM photogrammetry are known to have spatial uncertainties resulting from the different precision of each point [27,38]. Accounting for the (3D) precision of points of a point cloud by deriving detectable change has been highly recommended for geomorphic research publications [39]. They determine the actual levels of detection on a point basis which is crucial to not over-or underestimate topographic change [35,39]. A number of different approaches have been developed to derive precision estimates (1) by using the surface roughness as represented in the dense point cloud as a proxy for point precision (i.e., M3C2 plugin in CloudCompare [34,35]), (2) by accounting for random variations in the bundle adjustment performed within Agisoft PhotoScan running a Monte Carlo simulation on the sparse point cloud [38], or (3) by estimating the point precision by comparing the resulting point clouds of repeated unmanned aerial vehicle (UAV) surveys within a short timespan [27,40,41].

While the differences between SfM-MVS point clouds, TLS point clouds, and using different sensors for image acquisition for SfM-MVS photogrammetry have frequently been the focus of research [12,14,42,43], only few studies analyzed the effect of changing the perspective from terrestrial imagery to aerial imagery on the resulting point clouds and detectable changes [4,20,21,44]. Knowing the limitations and advantages of either approach is of high value for practitioners facing the challenge of documenting or monitoring small-to medium-scale geomorphic forms or landscapes (such as sinkholes, landslides, or riverbed changes) with limited equipment and time in the field.

Our objective of this study was the comparison of terrestrial and aerial (UAV) SfM photogrammetry of a sinkhole with complex terrain. We evaluate the resulting dense point clouds and detectable change while accounting for point precision estimates. Given the complex terrain and potential hazards of entering a sinkhole, we wanted to explore if terrestrial oblique images from around the edge of the sinkhole will result in a sufficiently

detailed and precise 3D point cloud to measure the changes within the sinkhole or if it is necessary to take aerial oblique images partly from within the sinkhole for improved coverage and point cloud quality. The terrestrial survey represents the easiest scenario of image collection for practitioners monitoring the sinkhole. As an alternative, we performed UAV flights during which the UAV was not only flying over the sinkhole (as usual in topographic surveys) but also within the sinkhole to acquire oblique imagery of the slopes and bottom of the sinkhole from up close, hypothesizing that a closer distance (range) and larger range of viewing angles will result in a higher-quality 3D reconstruction.

We selected an open sinkhole in northern Thuringia, close to the so-called "Äbtissinengrube", for our study as it provides a complex terrain with vegetated and unvegetated open slopes and it can be surveyed and approached from nearly every viewing angle by foot. Sinkholes are a major natural hazard in Thuringia (16,202 km$^2$; Germany), with more than 9000 sinkholes documented by the Geological Survey of Thuringia [45]. The surface dynamics of this open sinkhole are of particular interest to assess its stability. We surveyed the sinkhole once a year in 2017, 2018, and 2019. The images were taken terrestrially with a Nikon D3000 and aerially manually navigating a UAV (DJI Phantom Pro 4) equipped with a DJI FC330 (2017 and 2019) and DJI FC6310 (2018) camera. This long-term monitoring of changes of the sinkhole surface follows up on the work of [46].

We structured the paper as follows: we describe the study area, data collection, preprocessing, the 3D reconstruction, and the comparison of the resulting point clouds in the materials and methods section. The resulting point clouds, their precision maps, the derived distances between the point clouds, and the respective level of detection are presented in the results section. We provide a detailed discussion of our results within the light of other studies and other factors potentially influencing the quality of our point clouds in the discussion section and finally provide conclusions on our objective. All data collected and generated within this study are available for download at Zenodo [47].

## 2. Materials and Methods

### 2.1. Study Area

The sinkhole is located in an agricultural field in northern Thuringia, south of the Kyffhäuser Mountain Range, northwest of the town Bad Frankenhausen (Figure 1). It collapsed on the night of 16 to 17 November 2009 and has been enlarged by retrogressive erosion from a diameter of about 20 m (and 12–13 m depth [45,48]) to 24 m (own measurement 2019) ever since. Given its location, no remediation measures were taken, and the sinkhole is subject to erosion and vegetation growth. As sinkholes are a major natural hazard in Thuringia, the development and surface dynamics of this rare case of an open sinkhole are of interest for the local authorities.

With the presence of water (surface and groundwater) being a potential driver of sinkhole formation and, after its collapse, a driver of erosion processes, it is interesting to look at the yearly sum of precipitation at the study site. This area is comparably dry with a yearly average precipitation of 490 mm observed at the weather station Artern (1981–2010 [49]). During the years of our monitoring, the year 2017 experienced a lot more precipitation with 575 mm, while the year 2018 was very dry, with 273 mm. Even the observed precipitation sum of the year 2019 (399 mm) was well below the long-term yearly average [48]. In comparison, the yearly precipitation sum of the year of the sinkhole collapse (2009) was observed at 603 mm.

Geologically, the sinkhole is located in a transition zone between the "Diamantene Aue" flatland covered with loess, fluvial and glacial deposits, and the gypsum, anhydrite, dolostone, sandstone, and argillite of the "Kyffhäuser" mountain range. This transition zone is also referred to as the "Kyffhäuser Südrandstörung" (a suspected fault line) and experiences a high density of sinkholes of different ages (Figure 1A). In more detail, looking at the local stratigraphy, the following strata have been observed: (1) topsoil with sand and silt, potentially loess sediments, (2) starting at around five to twenty meters and reaching to about a maximum depth of 110 m the overlying sedimentary rocks of the Permian (so-called

"Zechstein"), and (3) starting from about 120 m depth sand and siltstone and conglomerates of the Permo-Carboniferous are forming the basement rocks [46]. The "Zechstein" is characterized by a sequence of good aquifers (e.g., dolostone, conglomerates) and aquitards (areas with anhydrite, loam, or schists with high clay content) and is well known for developing sinkholes evolving due to solution and collapse processes [45,46,50]. Within the sinkhole, some loose gypsum rocks and a thick layer of dark brown or black humus can be found on its northeastern slope, which might hint at an old, refilled depression [48]. Originally, most of the slopes formed straight walls (90° slope angle) which slowly experienced collapse or erosion, particularly at the western slope. Withstanding the erosion since the sinkhole occurrence, an earth pillar remained east of the center of the sinkhole as visible in the hillshade and photos of the sinkhole (Figure 1A,C–E). The local groundwater level is assumed to be at 50 m depth [45]. Unfortunately, more detailed information on the subsurface of the sinkhole (joints, groundwater flow, material, cavities) is not available.

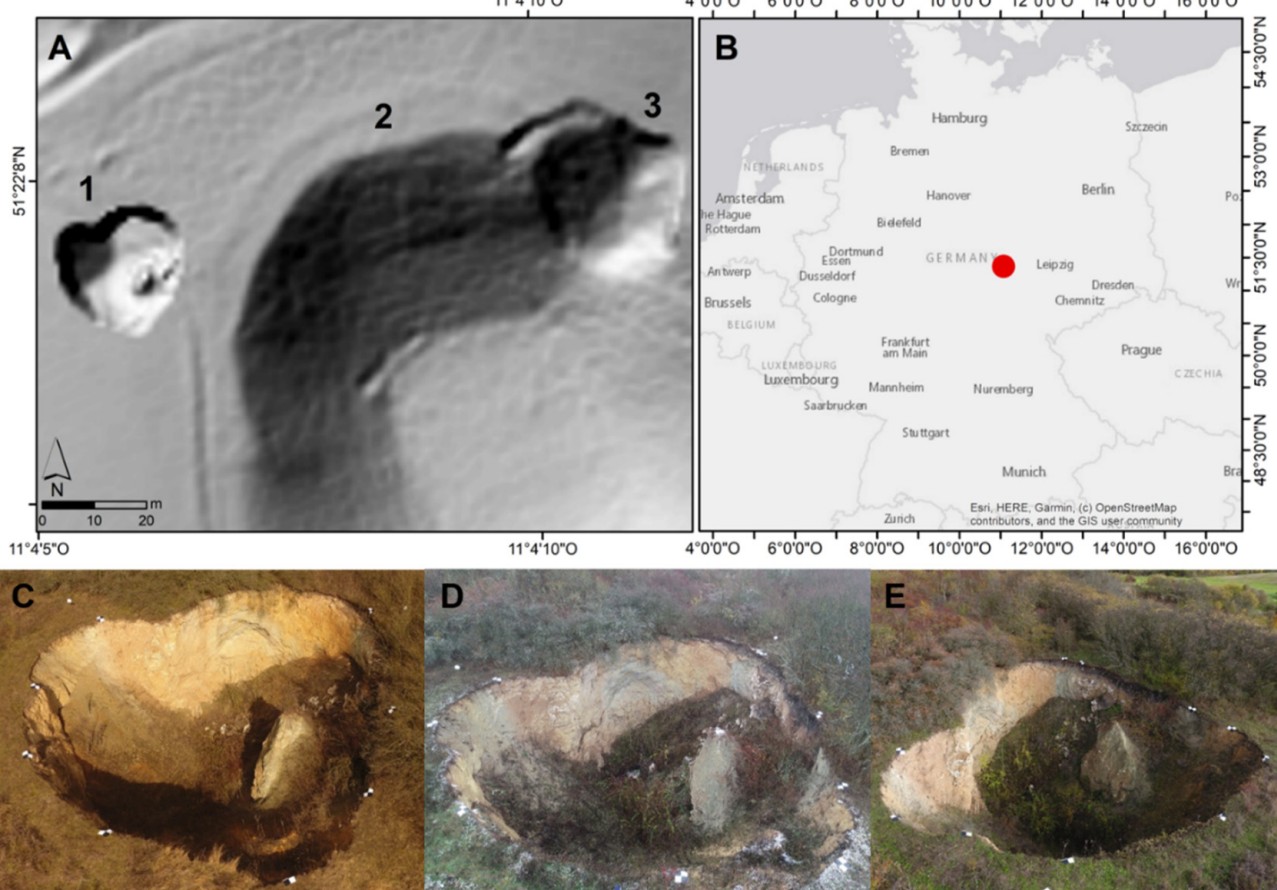

**Figure 1.** (**A**) (1) The studied sinkhole with the earth pillar east of the center, (2) the "Äbtissinengrube" of unknown age (myths date it back to 1600 A.D.), and (3) a sinkhole that occurred in 1953 at the edge of the "Äbtissinengrube". (**B**) Their location in Thuringia, Germany, marked with the red square. The sinkhole and survey conditions are captured as seen from the UAV in (**C**) 2017 facing north, (**D**) 2018 facing northeast, and (**E**) 2019 facing east. Please note the earth pillar and the white gypsum rocks at the bottom in the eastern half of the sinkhole. Source: (**A**) digital terrain model provided by the Thüringer Landesamt für Bodenmanagement und Geoinformation; (**B**) Esri; (**C–E**) photos taken by Jason Goetz in 2017, 2018, and 2019.

### 2.2. Data Collection and Preprocessing

We collected the data on 22 March 2017, 22 November 2018, and 5 November 2019 at the sinkhole. As visible in the photos of the sinkhole in Figure 1, the ground and weather conditions varied substantially between the three dates from sunny or slightly overcast

weather with dry slopes to foggy, overcast weather with some thin snow patches on the ground. Additionally, the vegetation cover within the sinkhole changed from no foliage and only dry stems in March 2017 to some foliage and increasingly growing vegetation in 2018 and 2019 (Table 1).

**Table 1.** Survey characteristics including the number of ground control points (GCPs).

| Date | Illumination Conditions | Ground and Weather Conditions | Number of GCP Targets |
|---|---|---|---|
| 22 March 2017 | Steep sun incidence, shadows visible. | Dry and brown soil, almost no foliage, dry stems; sunny, partly overcast, windy. | 8 |
| 22 November 2018 | Diffuse lighting, overcast. | Vegetation with green foliage, partly overgrown and snowy slopes; foggy with some snow. | 8 |
| 5 November 2019 | Steep sun incidence, shadows visible. | Vegetation with green foliage growing over the bottom and southeastern slope of the sinkhole; overcast, partly sunny, windy. | 10 |

The data collection and preprocessing consisted of three steps: (1) placing and surveying clearly visible targets (black and white checkered pattern) along the sinkhole edge and post-processing the GNSS survey; (2) acquiring terrestrial imagery with a handheld camera and converting the images to JPEG format; and (3) acquiring aerial imagery with an unmanned aerial vehicle (UAV; already JPEGs).

(1) Providing ground control points (GCPs) as a georeference for the resulting point clouds, we placed clearly visible targets (50 cm squares with a checkered pattern) along the edge of the sinkhole (as visible in Figure 1C–E). We placed the target center at the edge of the sinkhole with parts of the target reaching inwards to assure visibility from within and across the structure (8 targets in 2017 and 2018; 10 targets in 2019). Due to unknown hazards of a potential outlet at the bottom of the sinkhole and the potential of destroying parts of the sinkhole slopes, we did not enter the sinkhole to place any targets within. These GCPs were surveyed with the Leica Viva Global Navigation Satellite System (GNSS) base (GS10) and rover (GS15) RTK system before image acquisition. Manual post-processing was necessary, as the targets and base station were not in the same position each year. Therefore, the Sondershausen reference station was used applying post-processing kinematic (PPK) to our position data within the Leica Geo Office. We performed the study using the WGS89 UTM 32N reference system.

(2) We obtained terrestrial images using a Nikon D3000 DSLR handheld camera with a resolution of 3872 × 2592 pixels at 18 mm focal length. These oblique images were taken from the edge of the sinkhole with an overlap of approximately 60–80% between neighboring images. Due to dense vegetation in the north of the sinkhole, not all of the area surrounding the sinkhole was accessible by foot. The image-taking in 2017 and 2019 followed the suggestions of [51] of changing image position around the object, rather than changing viewing angle on one position. In 2018, the image-taking was carried out from a lower number of viewing points, from which images were taken with different viewing angles as in [33]. We converted the images from the Nikon raw format NEF to JPEGs with the Nikon Capture NX-D software applying the maximum-quality preset to prevent loss of detail.

(3) Oblique aerial images were taken with a DJI Phantom 4 Pro UAV equipped with a DJI FC330 sensor with 4000 × 3000 pixels at 4 mm focal length. Due to technical reasons, an FC6310 sensor with 5472 × 3648 pixels and 9 mm focal length was employed in 2018. The UAV was maneuvered manually to take images from within and above the sinkhole with a varying flight height of maximum 25 m. The images were taken regularly to ensure enough overlap in between neighboring ones.

### 2.3. Structure from Motion, Multiview Stereo 3D Reconstruction, and Computation of Precision Maps

The algorithms to create the tie points and dense point clouds from 2D images are implemented in various software and can be computationally demanding, depending on the number and quality of the images and the desired point cloud quality. Although there are open-source and free-of-charge options, the software Agisoft PhotoScan Pro (recently renamed Metashape, Version 1.4.3 from Agisoft LLC, St. Petersburg, Russia) is widely preferred in scientific applications because of the effective and fast implementation of the SfM-MVS algorithm [4] and a good quality of the resulting point clouds [42]. We used the version Agisoft PhotoScan Pro 1.4.3 (6529) to generate and reference the point clouds manually using markers on the targets. In the following, we describe each processing step in detail, stating the settings selected in Agisoft PhotoScan.

The images were loaded into separate PhotoScan projects for each year and sensor. Within PhotoScan, the images were manually examined again to exclude blurry, under- or overexposed ones from further analysis. We manually masked areas not portraying the sinkhole or obstructing vegetation within the sinkhole on the individual images in PhotoScan.

Next, we performed the camera alignment which worked fine for all the images except the images taken in 2018 terrestrially. We first noticed more than usual displaced points (sinkhole slopes appeared doubled or tripled); later on, the point precisions were magnitudes lower than for the other point clouds. Using coalignment after de Haas et al. [52] by including the 2018 UAV photographs for the camera alignment resolved this problem.

The SfM alignment process for the tie point detection was carried out with high accuracy, a limit of 40,000 key points, 10,000 tie points, masks applied to key points, and adaptive camera model fitting enabled. This process reconstructs camera positions and viewing angles on the basis of prominent features, referred to as tie points, within the images. The resulting sparse clouds were filtered gradually to only contain points of high quality in the reprojection error, reconstruction uncertainty, and projection accuracy. The sparse point filtering required balancing the removal of low-quality tie points for higher reconstruction quality, while also maintaining an adequate density of points to produce precision maps without any large data gaps [38]. The values were determined on a visual basis at 0.2, 15, and 10 for UAV and at 0.2, 30, and 15 for terrestrial imagery for the reprojection error, reconstruction uncertainty, and projection accuracy, respectively. The lowered reconstruction uncertainty threshold for the terrestrial imagery was chosen to ensure the sparse point clouds covered the sinkhole without too-large gaps. In between each step of gradual selection, the sparse point cloud's camera parameters were optimized. With this selection process we noticed that vegetated areas were greatly reduced within the sparse point cloud.

For georeferencing, the GCPs were imported and the center of the targets were manually identified and marked in the images. The UAVs' GPS and gyro-stored location and viewing angle for each acquisition were deemed inaccurate and disabled for alignment. Likewise, there was no geotagging for the images from the terrestrial handheld camera. The root mean square error (RMSE) of the marked GCPs was derived within Agisoft Photoscan Pro. We optimized the aligned sparse point cloud based on the GCP markers.

The multiview stereo 3D reconstruction was carried out by the creation of dense point clouds in PhotoScan based on the estimated camera parameters from the aligned images, applying mild-depth-filtering and medium quality. The dense clouds were exported in WGS 89 UTM 32N in "las" format for further processing in CloudCompare 2.12 beta (by Daniel Girardeau-Montaut is situated in Grenoble, France) [34].

We used the Monte Carlo simulation approach suggested by [38] to approximate the point coordinate precision of the dense point clouds. Using PhotoScan (V.1.4.5) and the software SfM_Georef 3.1, the coordinate precision (horizontal and vertical) of each point of the sparse point cloud is estimated by repeatedly optimizing the bundle adjustment. During each repetition, the positions of the ground control points are adjusted following a Gaussian distribution of the reported GNSS precision. The resulting sparse point cloud

precisions of each point include both SfM processing and georeferencing uncertainties. We estimated horizontal (*x* and *y* axis) and vertical (*z* axis) precisions ($\sigma$XYZ) of each point in the georeferenced sparse clouds using a 10,000-fold Monte Carlo simulation [38]. Following the procedure by [38], we applied a distance-based spherical normal distribution interpolation in CloudCompare to interpolate the precision estimates from each sparse to the respective dense point cloud.

### 2.4. Point Cloud Comparison and Deformation Analysis

We analyzed differences in the point clouds using terrestrial or aerial imagery and surface changes within the sinkhole, comparing the dense point clouds, considering their point precisions using the multiscale model to model cloud comparison with precision maps plug-in (M3C2-PM [35]) within CloudCompare 2.12 beta.

The application of the M3C2-PM plugin required two preprocessing steps within CloudCompare. First, the proper coregistration of the dense point clouds needed to be assured. As each survey was georeferenced manually using high-precision post-processed GNSS data in PhotoScan and the georeferencing precision is integral in the precision maps we derived by SfM_Georef, no further registration was carried out between the point clouds. Second, the M3C2-PM plugin requires normals of each point of the dense point cloud to determine the distance between two point clouds. The M3C2 algorithm creates cylinders around the core points aligned to the local normals, inside of which the mean distances of the points of the source cloud to the target cloud are measured. Reportedly, the correct orientation of the normals has a large influence on correctly portraying the distances between the point clouds [35]. The normal scale describes the size of the spherical neighborhood around each point used to compute its local normal. Depending on the complexity and size of the analyzed terrain and the available computation time, Lague et al. [35] recommend the computation of multiscale normals instead of applying a uniform normal scale for the entire point cloud. Their proposed method of deriving multiscale normals tests a range of normal scales on each core point, instead of one uniform normal scale, to identify the scale with the most planar resulting normal for each core point. In this way, the normals of rough edges and more smooth planar surfaces can be portrayed with equally suitable normals and the overestimation of the measured distance between the point clouds is avoided (for more details and discussion of this approach please refer to [35]). We applied the multiscale normal calculation within the M3C2 plugin starting with a minimum scale of 0.1 m and ending at a maximum scale of 2.6 m using steps of 0.5 m. This represents the recommended 20–25 times-as-large normal scale compared to the local mean relative surface roughness, which was found to be around 0.1 in our point clouds. We selected the preferred orientation of the normals as towards the barycenter of the sinkhole, which worked for most normals. However, due to the complex shape of the sinkhole, we had to apply a minimum spanning tree with a maximum of 15 neighbors after the multiscale normal calculation to orient all normals correctly facing inwards the sinkhole.

We calculated the distance between the point clouds using the correctly oriented multiscale normals and setting the diameter of the cylinder (called projection scale) at 0.2 m and the maximum cylinder depth at 1 m. These values were found to be appropriate, as we did not expect any reliably measured distances to be larger than that. Within each comparison pair we used the entire point cloud with the largest amount of points as the reference cloud without applying any sampling of core points. In case the reference cloud was younger than the comparison cloud, we inverted the measured distances to match the change direction in other comparison pairs. Including the precision maps in the M3C2 analysis allows to account for local uncertainties, introduced by the quality of GPS measurements and the 3D reconstruction. It provides a more detailed view on actual detectable change, taking into account the local level of detection of each point which summarizes the distance uncertainty provided by the precision maps for each point [35,38]. The result is a point cloud containing information on the measured distance (m) and local level of detection (m), and combining these on points with detectable change (named M3C2

distance, distance uncertainty, and significant change within the plugin). Detectable change refers to the points where the local uncertainty (or level of detection) is lower than the measured change.

Comparisons were carried out between terrestrial and aerial point clouds of the same year (2017, 2018, and 2019 respectively) to identify strengths, weaknesses, and the general point cloud quality of each data collection approach. Additionally, we were interested in the detectable surface changes or occurred deformation within the sinkhole over the years conveyed by each data collection method. Therefore, we compared the point clouds over one year (2017 to 2018 and 2018 to 2019) and two years (2017 to 2019) using the same data collection approach (terrestrial or aerial). This resulted in nine comparison pairs, as presented in the results section.

As we had no independent validation dataset from another source, such as point surveys from a differential GPS, a total station, or a TLS survey, available, we assessed the quality of the resulting point clouds according to these criteria: the amount of points in the sparse and dense point cloud, the (median) point density as described by the number of neighbors at the M3C2 projection scale (0.2 m), general coverage of the sinkhole (the size of holes in the point cloud were assessed by deriving the percentage of black pixel in a rendered file (PNG) of the each point cloud viewed from the top), the mean precision estimate ($sX$, $sY$, and $sZ$) and its standard deviation, and observed change (analyzing if change was measured differently in particular parts of the sinkhole using the terrestrial or aerial point cloud).

## 3. Results

### 3.1. Data Collection and Preprocessing

The resulting median GNSS survey accuracy (positional and height errors) of the surveyed ground control points varied from 1 cm (at 7 mm stdev. of height) in 2017 to 1.8 cm and 2 cm (at 16 mm or 18 mm stdev. of height) in 2018 and 2019, respectively.

After filtering the data for blurry or out-of-focus images, we count more images taken with the UAV (up to 564 images) than with the handheld camera (as few as 94 images in 2017, as summarized in Table 2). The terrestrial images were acquired from rather similar locations around the sinkhole except from its north, as dense bushes make this area inaccessible by foot. The camera locations of the aerial images differ substantially in between the years as a result of the manual flight mode. Nonetheless, a good image overlap of more than nine images was achieved with each sensor (please refer to maps showing the camera locations and image overlap in Appendix A, Figure A1).

**Table 2.** Overview of sensors and data/images.

| Sensor | Resolution | Focal Length | Number of Images | | |
|---|---|---|---|---|---|
| | | | **2017** | **2018** | **2019** |
| Nikon D3000 | 3872 × 2592 | 18 mm | 94 | 166 | 178 |
| DJI FC330 | 4000 × 3000 | 4 mm | 353 | - | 287 |
| DJI FC6310 | 5472 × 3647 | 9 mm | - | 564 | - |

### 3.2. Structure from Motion, Multiview Stereo 3D Reconstruction, and Computation of Precision Maps

The number of points in the sparse clouds is directly influenced by the gradual filtering. The year 2018 shows both the lowest amount of sparse cloud points in the terrestrial dataset (6326 points) and the most points in sparse point cloud resulting from using the UAV images (54,443 points; Table 3). The dense point clouds are similar in size, apart from the 2018 UAV dense point cloud, which shows almost three times the amount of points compared to the others (8767,159 points, Table 3; the gradual filtering was the same for all UAV surveys, though). Comparably, the mean point density giving the number of neighbors within a 0.2 m sphere of the dense point clouds is highest for the point cloud resulting from the 2018 aerial imagery (359 neighbors/0.2 m). However, the point density

is not lowest for the point cloud with the lowest number of points, as it is lowest for the 2017 terrestrial point cloud (101 neighbors/0.2 m compared to 108 neighbors/0.2 m in the 2018 terrestrial point cloud). Except for the 2018 data, the point density is comparable between point clouds resulting from terrestrial and aerial imagery. Notably, the point density does not describe the coverage of the sinkhole in general well. Therefore, we assessed the percentage of black pixels in the picture (PNG) viewing the sinkhole from the top. It is the lowest for the point clouds resulting from the aerial imagery (51% in 2017 and 2018, 52% in 2019; Table 3), indicating a very good coverage of the sinkhole. In the reconstructed dense point clouds, the coverage of unvegetated areas is mostly very good. The largest holes were found in the point cloud resulting from the 2018 terrestrial imagery (70% black pixels), where points in a large area of the sinkhole floor and area behind the earth pillar are missing. The holes in the other point clouds are mostly related to vegetated areas on the sinkhole bottom and behind the earth pillar.

**Table 3.** Characteristics of the resulting point clouds per year and survey method. The mean point density was evaluated on the dense point cloud at the projection scale of the M3C2 method at 0.2 m. The coverage was assessed by deriving the percentage of black pixels in a PNG format image of each dense point cloud from the top. Please note that the percentage does not portray the actual size of the holes, as the surroundings of the sinkhole are also in black. The PNGs can be found in the Appendix A Figure A2a–e.

| Year | Survey Method | Number of Images Used | No. of Points Sparse Cloud | No. of Points Dense Cloud | Control Points RMSE * (mm) | Mean Point Density (No. of Neighbors/ 0.2 m) | Coverage (% of Black Pixels) | Point Precision Estimates ($\sigma_X$) (mm) | | Point Precision Estimates ($\sigma_Y$) (mm) | | Point Precision Estimates ($\sigma_Z$) (mm) | |
|---|---|---|---|---|---|---|---|---|---|---|---|---|---|
| | | | | | | | | Mean | Std. dev. | Mean | Std. dev. | Mean | Std. dev. |
| 2017 | Aerial | 353 | 37,037 | 2663,766 | 12 | 101 | 51 | 3 | 0.5 | 4 | 0.5 | 5 | 1.1 |
| | Terrestrial | 94 | 13,051 | 2,487,248 | 35 | 104 | 58 | 25 | 15.1 | 28 | 14.3 | 15 | 8.1 |
| 2018 | Aerial | 542 | 54,443 | 8,767,159 | 15 | 359 | 51 | 10 | 4.8 | 12 | 6.2 | 17 | 6.8 |
| | Terrestrial | 165 | 6326 | 2,267,846 | 25 | 108 | 70 | 20 | 10.1 | 18 | 10.1 | 13 | 3.2 |
| 2019 | Aerial | 278 | 34,612 | 2,673,191 | 12 | 109 | 52 | 6 | 2.5 | 7 | 2.3 | 16 | 8.2 |
| | Terrestrial | 177 | 21,929 | 2,707,662 | 17 | 117 | 65 | 25 | 12.8 | 25 | 12.2 | 16 | 5.9 |

\* RMSE: root mean square error.

Looking at the RMSE of manually marking the GCPs in the images, the highest RMSE resulted in the point cloud of the 2017 terrestrial imagery (35 mm). Using the same GCP survey but still manually marking their location in the 2017 aerial photos resulted in the lowest RMSE of 12 mm. This very low RMSE was also found for the GCPs of the 2019 aerial images.

The mean horizontal point precision estimates ($\sigma X$ and $\sigma Y$) are generally higher for the point clouds resulting from terrestrial imagery ranging from 18 mm ($\sigma Y$ of 2018 terrestrial) to 28 mm ($\sigma X$ of 2017 terrestrial; Table 3). The vertical point coordinate precision ($\sigma Z$) does not share these differences as the precision estimates are ranging between 13 mm and 17 mm for all surveys, with the exception of the point cloud resulting from the 2017 aerial survey (5 mm; Figure 2). Looking at the spatial distribution of the vertical point coordinate precisions from the terrestrial surveys, a consistent general pattern appears for all years (Figure 2): The center of the sinkhole, which is furthest from the GCPs, displays larger variations within the bundle adjustment, resulting in lower precision of the point coordinates. The largest precision estimate values in the brown colored areas (starting at a $\sigma Z$ of 30 mm) are attributed to vegetation within the images (see photos in Figure 1). The slopes display a lower degree of variation in the vertical point precisions, ranging between 5 mm and 15 mm. The vertical precision maps of the aerial imagery show different patterns in all surveys. The 2017 aerial point cloud has excellent vertical point coordinate precisions (with a maximum value of $\sigma Z = 15$ mm) with only minor deviations at the center and north part of the sinkhole (Figure 2). The 2018 aerial point cloud displays the largest vertical precision $\sigma Z$ of up to 60 mm. The precision map of the 2019 aerial point cloud shows a distinct gradient of larger vertical precision values to the northeast, while the precision is generally good ($\sigma Z$ of 6 mm to 33 mm).

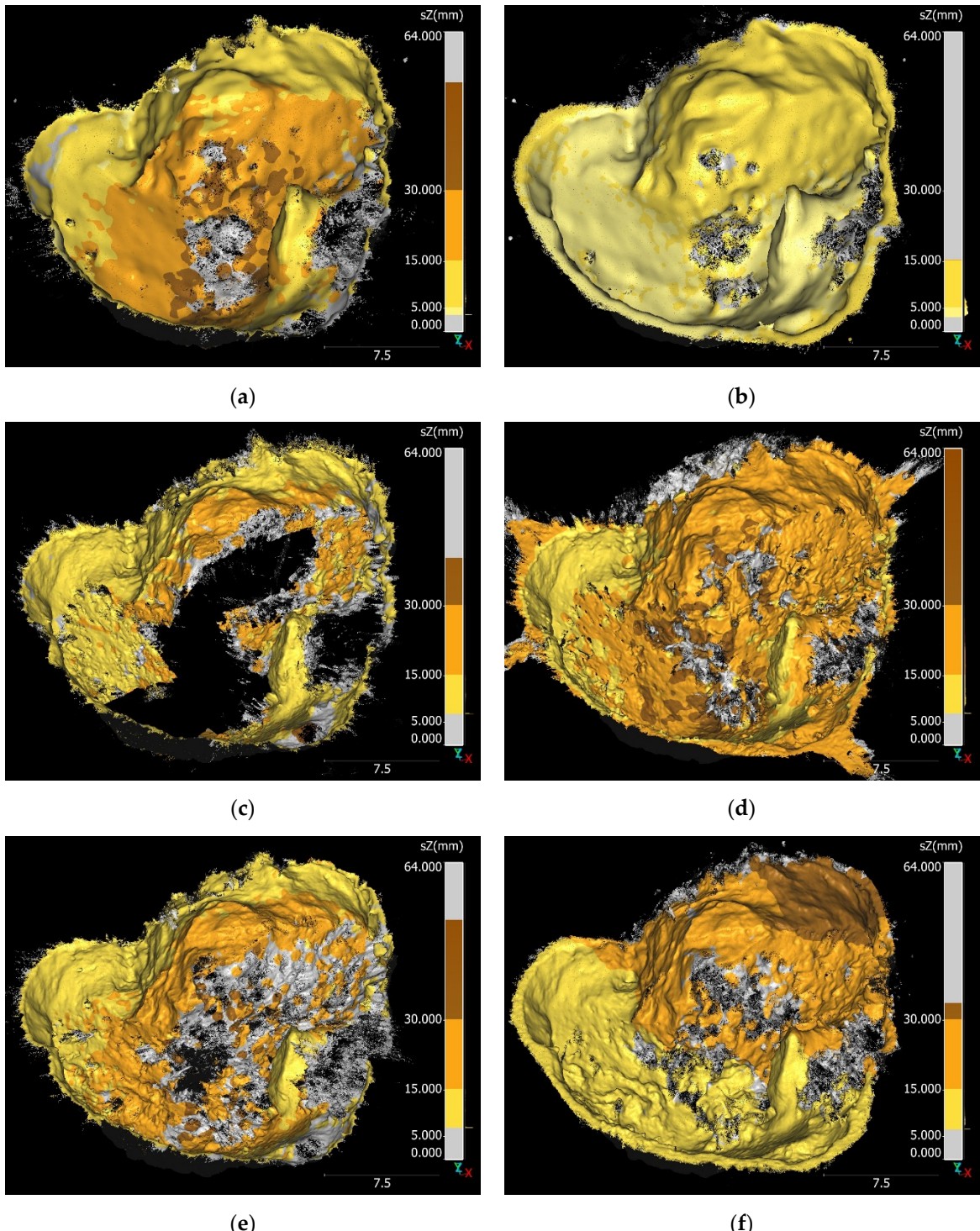

**Figure 2.** Vertical precision maps for the point clouds of the 2017 to 2019 terrestrial (**a**) Terrestrial 2017 precision map ($\sigma Z$), (**c**) Terrestrial 2018 precision map ($\sigma Z$), (**e**) Terrestrial 2019 precision map ($\sigma Z$) and aerial (**b**) Aerial 2017 precision map ($\sigma Z$), (**d**) Aerial 2018 precision map ($\sigma Z$), (**f**) Aerial 2019 precision map ($\sigma Z$) imagery. Gray color indicates areas where no point precision estimate was available from the interpolation from the sparse point cloud. Note that in these areas, also, no detectable change can be derived as no local level of detection is available. Additionally, please note the different minimum and maximum precision estimate value of each point cloud as indicated by the colors shown in the color scale.

### 3.3. Point Cloud Comparison and Deformation Analysis

Comparing the point clouds resulting from using the different sensors, the percentage of points with detectable change is a very interesting metric to look at, as it gives us the points which show differences between the point clouds that exceed the precision uncertainty of each point cloud. Any detectable changes or differences between the point clouds of the same year are likely effects from the imagery and camera locations, as well as the different sensors. The agreement between the terrestrial and aerial point clouds of 2018 and 2019 is considered high, given that only 14% (2018) and 11% (2019) of the points have detectable change (Table 4). The mean measured distance between these point clouds is also rather small, with 3 mm or −11 mm (all points or points with detectable change in 2018) and 8 mm or 19 mm (all points or points with detectable change in 2019). However, the comparison of the 2017 terrestrial and aerial point clouds shows detectable change for 43% of all points of the point cloud resulting from aerial imagery, with a mean measured distance of 40 mm on the entire point cloud and of 78 mm for the points with detectable change (Table 4).

**Table 4.** Results of the M3C2 comparisons made between the terrestrial and aerial point clouds to analyze the difference between using a different sensor and between the years to analyze the changes within the sinkhole.

| Comparison ID | Reference Cloud | Compared Cloud | Mean Measured Distance (mm) | | Mean Local Level of Detection (mm) | | % of Reference Cloud Points with Detectable Change |
|---|---|---|---|---|---|---|---|
| | | | All Points | Points with Detectable Change | All Points | Points with Detectable Change | |
| TerrUAV 2017 | 2017 U [1] | 2017 T [2] | 40 | 78 | 54 | 47 | 43 |
| TerrUAV 2018 | 2018 U | 2018 T * | 3 | −11 | 48 | 44 | 14 |
| TerrUAV 2019 | 2019 U | 2019 T | 8 | 19 | 55 | 51 | 11 |
| Terr 2017/18 | 2017 T | 2018 T * | −49 | −103 | 67 | 56 | 33 |
| Terr 2018/19 | 2019 T | 2018 T * | 58 | 136 | 64 | 59 | 25 |
| Terr 2017/19 | 2019 T | 2017 T | 25 | 20 | 73 | 65 | 30 |
| UAV 2017/18 | 2018 U | 2017 U | −27 | −39 | 32 | 30 | 52 |
| UAV 2018/19 | 2018 U | 2019 U | 61 | 101 | 36 | 35 | 38 |
| UAV 2017/19 | 2019 U | 2017 U | 37 | 20 | 21 | 65 | 62 |

[1] Terrestrial (T); [2] UAV (U); * Coregistered.

Spatially, the areas of detectable change can mainly be found on the partly vegetated sinkhole floor, while most of the steep slopes (unvegetated, except the western slope) usually show a good agreement between the point clouds resulting from different sensors (Figure 3). Please note that detectable change can only be assigned in areas where both point clouds provide precision estimates and points. As the point cloud resulting from the 2018 terrestrial imagery has a large area on the sinkhole floor with no points and no precision estimates, no detectable change can be assigned there, which may lead to artificially improved results of the percentage of points with detectable change. In all years, the largest measured distances (larger than ±15 cm) can be found in vegetated areas (Figure 3). In 2017, the rather large measured distances of 5 cm to 15 cm on the sinkhole floor stand out compared to the results of the other years. The comparison of the sensors in 2018 shows more areas of detectable change at the northwestern slope of the sinkhole, where the structure reconstruction from the UAV has a higher surface (measured distances of −5 cm to −15 cm) (Figure 1c,d). The negative change is locally disrupted by circular blobs with measured distances larger than 15 cm, where the terrestrial point cloud is locally higher than the aerial one. These areas coincide with the more abundant vegetation in 2018, especially stalks. Dominant are areas with the missing points at the sinkhole floor (in gray) and the vegetated area southeast of the earth pillar, which displayed too-low coverage in the terrestrial images (Figure 3). In 2019, the pattern of change between the sensors is visibly more complex, with a measured distance of most of the points between ±5 cm (Figure 3). The aforementioned blobs of points in purple and dark blue (±15 cm) are also present here, particularly across the sinkhole floor and western slope.

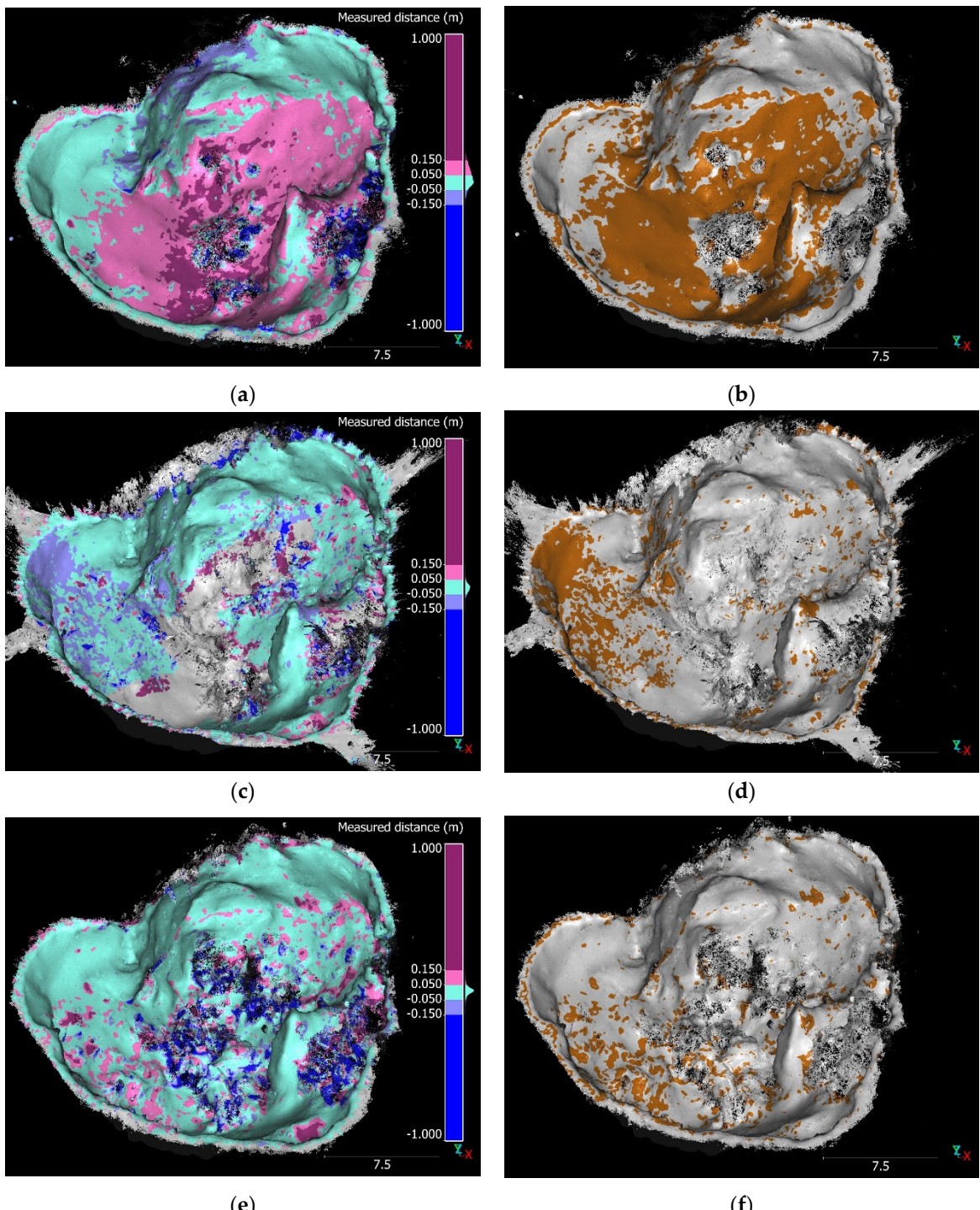

**Figure 3.** Measured distances and points with detectable change (brown) for comparisons of TerrUAV 2017, 2018, and 2019: (**a**) TerrUAV 2017 measured distance (m) between terrestrial and UAV point cloud 2017, displayed on the UAV point cloud. (**b**) TerrUAV 2017 detectable change between terrestrial and UAV point cloud 2017. (**c**) TerrUAV 2018 measured distance (m) between terrestrial and UAV point cloud 2018, displayed on the UAV point cloud. (**d**) TerrUAV 2018 detectable change between terrestrial and UAV point cloud 2018. (**e**) TerrUAV 2019 measured distance (m) between terrestrial and UAV point cloud 2019, displayed on the UAV point cloud. (**f**) TerrUAV 2019 detectable change between terrestrial and UAV point cloud 2019. Note the different information portrayed by the color gray: in (**a**,**c**,**e**), gray areas show areas where no change was measured due to missing points in one of the point clouds; in (**b**,**d**,**f**), the gray color shows points with no detectable change.

Additionally, we were interested in the surface changes over time portrayed by each surveying method. Using the terrestrial point clouds from 2017 to 2018, large portions of the sinkhole are showing negative detectable change of larger than −5 cm within the western slope and even larger than −15 cm visible at the earth pillar (Figure 4). The mean measured distance of all points is −49 mm (or −103 mm for points with detectable change), with 33% of all points of the 2017 terrestrial point cloud resulting in detectable change (Table 4). The northern wall mainly shows changes of ±5 cm, which are, for the most part, not reflected within the detectable change. The areas of detectable change there have measured distances larger than −15 mm. The southern slope shows a positive detectable change of 5 cm to 15 cm, which can be attributed to vegetation growth. Additionally, small areas of negative change are detectable at the edge around the sinkhole. In contrast, the comparison of terrestrial point clouds from 2018 to 2019 shows a positive change of 5 cm and larger than 15 cm overall, with only a minority of negative measured distances. The mean measured distance for all points is also positive, with 58 mm (136 mm for points with detectable change) with a mean local level of detection of 64 mm (Table 4). The detectable change (25% of all points of the 2019 terrestrial point cloud) shows a similar pattern to the 2017/18 comparison: the difference between the point clouds is detectable within the western slope and not detectable in the northern wall. The top of the earth pillar again shows detectable but smaller areas of negative change (between minus 5 cm and minus 15 cm; Figure 4). The change detection over the full timespan from 2017 to 2019 using terrestrial point clouds displays patterns of both single-year comparisons: most notably, the detectable negative change at the earth pillar (larger than minus 15 cm) and the positive change at the less steep parts of the western slope (patches of change larger than 15 cm) and the steep southern slope (more than 15 cm). In addition, the negative change at the edge of the sinkhole, which was not visible from 2018 to 2019, is present. The mean measured distance in this comparison pair is 25 mm for all points (20 mm for points with detectable change) at a mean local level of detection of 73 mm. A total of 30% of the points of the 2019 terrestrial point cloud were found to have measured distances exceeding the local level of detection, resulting in detectable change.

The change detection over time using the point clouds from the aerial (UAV) imagery resulted in partly similar patterns and mean measured distances compared to the changes found in the point clouds from the terrestrial imagery, with generally lower mean local levels of detection and larger areas of detectable change: The 2017/18 comparison shows detectable negative change at the earth pillar (but smaller areas with distances larger than −15 mm) and at the upper part of the western slope (−15 mm to 5 mm), and shorter measured distances in areas of detectable change in the northern wall (mainly ±5 mm and only small areas of −15 mm to −5 mm). The large areas of negative detectable change at the bottom of the western slope are missing. Only the positive blobs of the stalks at the transition of the steep western slope to the flatter lower part are also visible at the same measured distance in the comparison of the aerial point clouds of 2017/18. The mean measured distance for all points is −27 mm (−39 mm for points with detectable change) at a mean local level of detection of 32 mm. A total of 52% of the 2018 aerial point cloud resulted in detectable change. From 2018 to 2019, the point clouds resulting from aerial imagery show more measured distances at the western and northern slope within ±5 mm and similar detectable change on the western and eastern sinkhole floor (larger than 5 mm and 15 mm). The negative change at the stalks is also visible at the same measured distance (larger than −15 mm). The distances measured at the earth pillar are comparable to the terrestrial result, range within −5 mm and −15 mm, and are considered detectable. The mean measured distance is 61 mm for all points (101 mm for the points with detectable change) and the mean local level of detection is 36 mm, which resulted in 38% of the 2018 aerial point cloud showing detectable change. The two-year comparison (2017 to 2019) gives a clearer image of the changes, and the results of this comparison pair are also most similar to the results of using the terrestrial imagery for the point cloud computation. We found negative detectable measured distances between −5 mm and −15 mm at the upper

part of the western slope and larger than −15 mm at the earth pillar, comparable to the results of the terrestrial point clouds (Figure 5). Positive detectable measured distances (larger than 5 mm) at the lower part of the western slope, the southern slope, and the bottom of the sinkhole are also comparable. However, there are some differences, most strikingly at the sinkhole floor north of the earth pillar, where the terrestrial point clouds (2017/19) showed negative change larger than −15 mm while the aerial point clouds (2017/19) mainly showed positive change. The mean measured distance between the aerial point clouds of 2017 and 2019 is 37 mm for all points (20 mm for points with detectable change) and the mean local level of detection is 21 mm. This comparison pair resulted in the largest area of detectable change, with 62% of the points of the 2019 aerial point cloud.

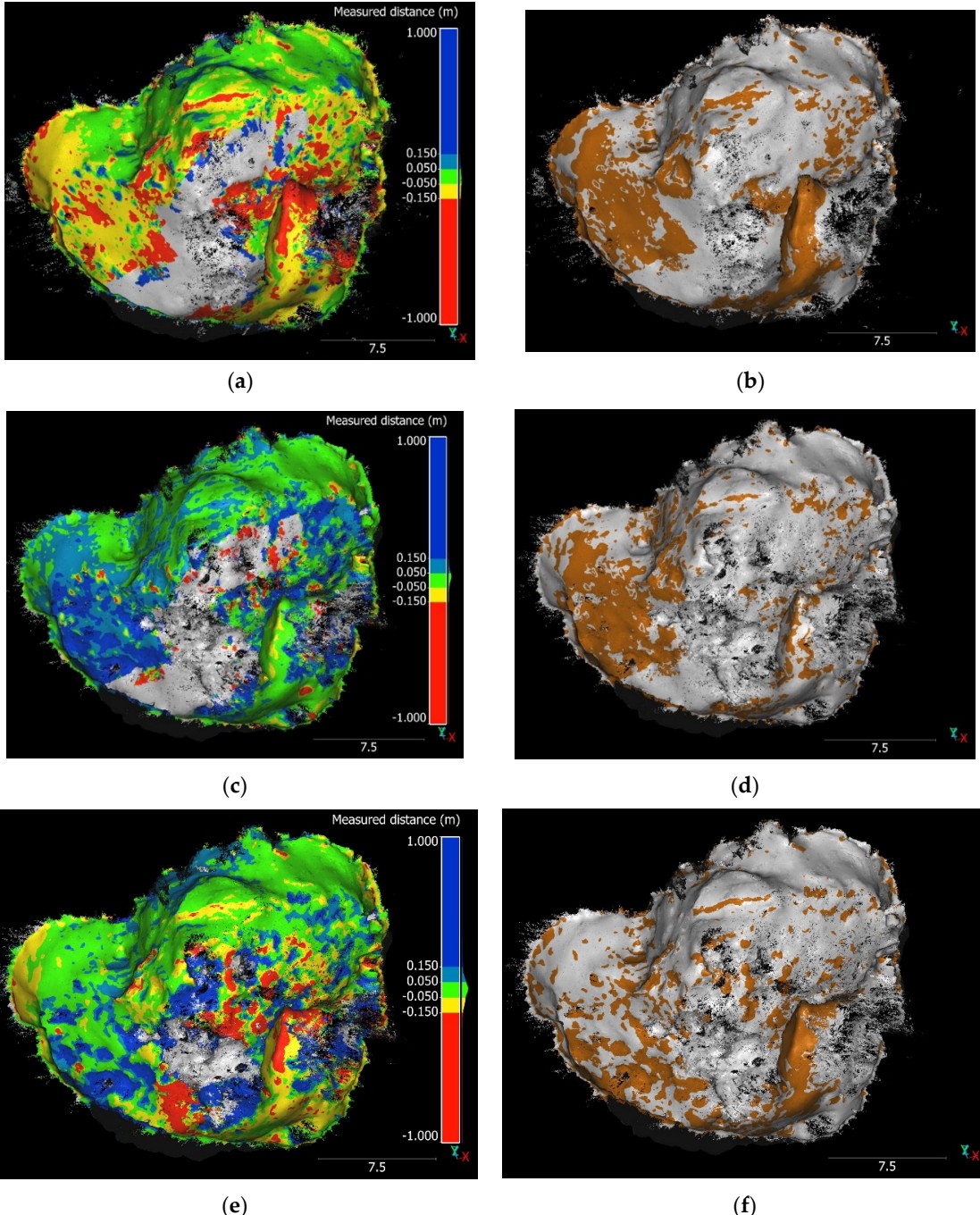

**Figure 4.** Measured distances and regions of detectable change for comparisons Terr 2017/18, 2018/19, and 2017/19 (see Table 4): (**a**) Terr 2017/18 measured distance (m), Terr 2017/18: measured distances

of terrestrial 2017 to terrestrial 2018. (**b**) Terr 2017/18 detectable change, Terr 2017/18: detectable change of terrestrial 2017 to terrestrial 2018. (**c**) Terr 2018/19 measured distance (m), Terr 2018/19: M3C2 distances of terrestrial 2018 to terrestrial 2019. (**d**) Terr 2018/19 detectable change, Terr 2018/19: detectable change of terrestrial 2018 to terrestrial 2019. (**e**) Terr 2017/19 measured distance (m), Terr 2017/19: measured distances of terrestrial 2017 to terrestrial 2019. (**f**) Terr 2017/19 detectable change, Terr 2017/19: detectable change of terrestrial 2017 to terrestrial 2019. Note the different information portrayed by the color gray: in (**a**,**c**,**e**), gray areas show areas where no change was measured (due to missing points or the maximum depth of 1 m was exceeded); in (**b**,**d**,**f**), the gray color shows points with no detectable change.

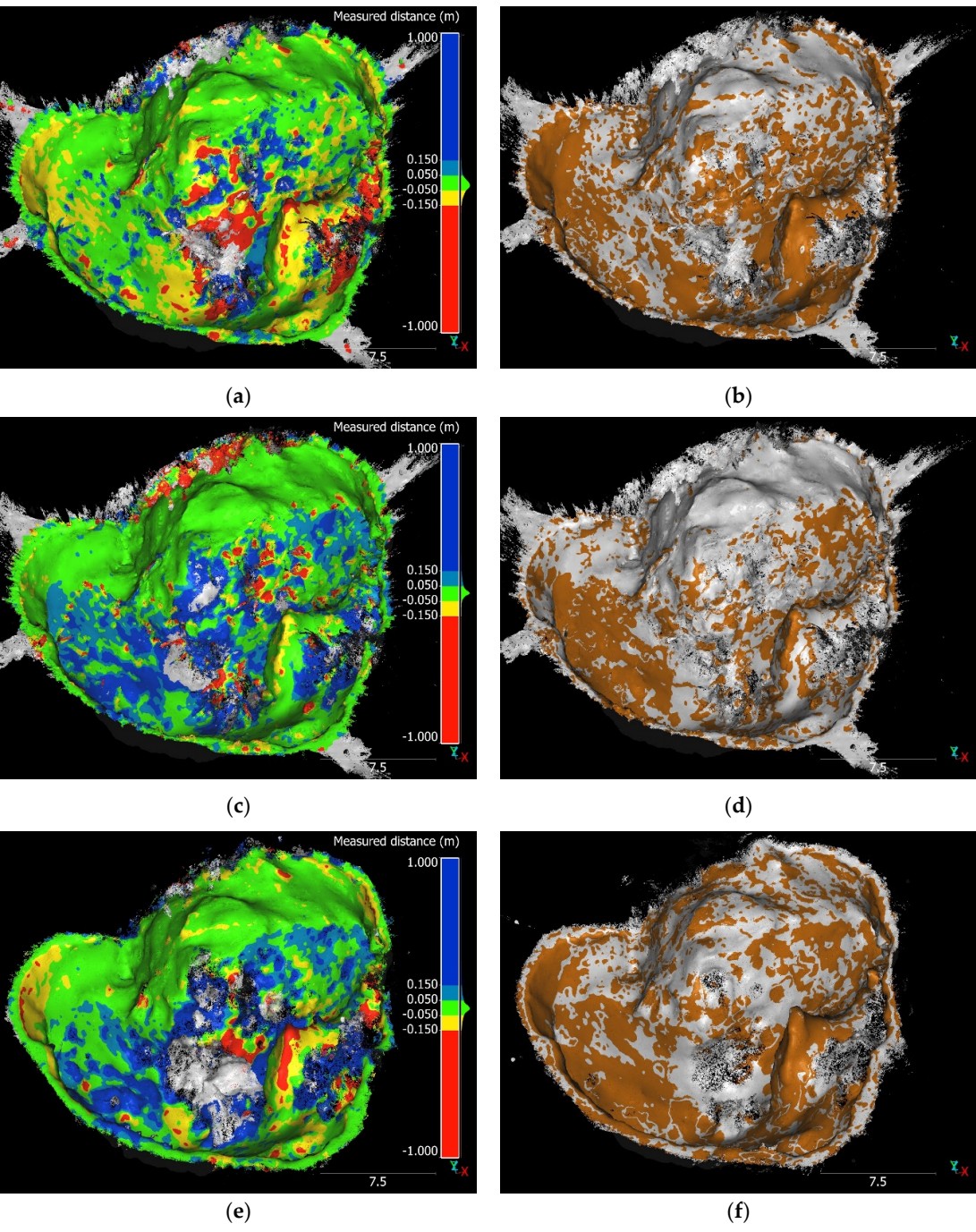

(**a**)

(**b**)

(**c**)

(**d**)

(**e**)

(**f**)

**Figure 5.** Measured distances and regions of detectable change for comparisons UAV 2017/18, 2018/19, and 2017/19 (see Table 4): (**a**) UAV 2017/18 measured distance (m), UAV 2017/18: measured

distances of UAV 2017 to UAV 2018. (**b**) UAV 2017/18 detectable change, UAV 2017/18: detectable change of UAV 2017 to UAV 2018. (**c**) UAV 2018/19 measured distance (m), UAV 2018/19: measured distances of UAV 2018 to UAV 2019. (**d**) UAV 2018/19 detectable change, UAV 2018/19: detectable change of UAV 2018 to UAV 2019. (**e**) UAV 2017/19 measured distance (m), UAV 2017/19: measured distances of UAV 2017 to UAV 2019. (**f**) UAV 2017/19 detectable change, UAV 2017/19: detectable change of UAV 2017 to UAV 2019. Note the different information portrayed by the color gray: in (**a**,**c**,**e**), gray areas show areas where no change was measured (due to missing points or the maximum depth of 1 m was exceeded); in (**b**,**d**,**f**), the gray color shows points with no detectable change.

## 4. Discussion

### 4.1. Comparing Terrestrial and UAV Patterns of Change

Looking at the results of comparing the point clouds resulting from terrestrial or aerial imagery from the same year, the sometimes small (2019) or larger differences (2017) between the point clouds are to some extent surprising. As this is a comparison of sensors with data taken at the same day within minutes of each other, we can safely assume no change of the sinkhole happened between acquisition of the aerial and terrestrial imagery. All differences measured here are produced by the different sensors, perspective, camera locations, and SfM and MVS processes. Particularly, the differences in the 2017 point cloud pair showing large differences on the sinkhole floor are interesting. We assume the higher position may be attributed to the limited viewing angles of the terrestrial camera resulting in a bad image quality in areas of the sinkhole floor. It could also be related to some sort of doming effect resulting from poor camera and ground control point positioning. These errors are explicitly not portrayed in the point precision maps [5]. However, with the 2019 terrestrial survey design being very similar, this different representation of the sinkhole floor did not happen then. Therefore, the exact reason for the difference remains unclear. The small but detectable differences between the point clouds resulting from the two sensors in 2019 show a pattern that may be attributed to point precision variations resulting from the randomness during photogrammetry 3D reconstruction [38,53,54] and/or to differences in the reconstruction of the vegetation on the sinkhole floor (such as due to gradual filtering or vegetation movement).

With the multitemporal comparison using the same survey method but point clouds of different years, we had different challenges. While we have no means of knowing which measured distances portray the real change within the sinkhole over time, we can apply general geomorphological knowledge and principles to evaluate the validity or plausibility of the changes we measured. Additionally, it is beneficial to only look at mainly vegetation-free areas or areas with only very low vegetation. In these areas, the influences of the gradual filtering tool and the vegetation movement and growth during and in between the surveys are assumed to play no role in the point cloud quality or the spatial pattern of measured distances. We also assume that changes we see in both terrestrial and aerial comparison pairs of the respective years might be real changes (erosion, deposition, or vegetation growth) that occurred within the sinkhole. Looking at the multitemporal point cloud comparison results of comparison pairs, using the 2018 terrestrial point cloud seems least geomorphologically plausible.

While the patterns of change detected within point clouds resulting from terrestrial or aerial imagery appeared quite different whenever the 2018 terrestrial point cloud was involved, the patterns of change found between the years 2017 and 2019 (particularly the erosion at the earth pillar and western steep slope and deposition at the lower western slope) are comparable. The representation quality seems to have been very good at the earth pillar in all years and we can see that in the first year (2017 to 2018), when a lot of precipitation was recorded, the largest amount of erosion took place at the top of the earth pillar. From 2018 to 2019 there was still some erosion at the top but much less, which fits very well with the low recorded precipitation in those years. Depending on the aimed-for level of detection or measured distances with detectable change and the main surveyed area

(slope or sinkhole floor), the choice of preferable survey method might be different. Our results confirm that similar patterns of change can be detected using either survey method at the cost of higher (terrestrial) or lower (aerial) local levels of detection and smaller areas of detectable change (terrestrial), if the survey itself is performed properly. We assume that the slightly different approach of larger distances between the camera locations along the sinkhole edge and orientation towards the sinkhole in the terrestrial survey 2018 (with larger angular changes in between the single images; Figure A1) is what caused the trouble with the alignment (which is why the coalignment was necessary) and resulted in a point cloud with very large areas of the sinkhole (floor) missing. Angular changes greater than 25–30° between images have been found to be troubling for feature detectors [55]. This finding of not-optimal image acquisition methods applied with the handheld camera in 2018 is supported by survey method recommendations summarized by Bemis et al. [20], James and Robson [56], and Lin et al. [57]. Therefore, we strongly advise against the 2018 terrestrial survey approach and against overinterpreting the results which involve this point cloud, particularly as the large areas of change on the western slope (negative change in the comparison pair Terrestrial 2017/18 and positive change in the comparison pair Terrestrial 2018/19) can be attributed to the 2018 terrestrial point cloud (as it is not visible in the respective aerial results) and do not seem plausible from a geomorphological point of view.

The difference between the range and spatial coverage of the local level of detection for terrestrial and aerial point clouds is striking, making the aerial point clouds more appealing when trying to achieve centimeter-precision surveys. The spatial coverage is a result of the presence of information on point precision estimates in both point clouds of the comparison pair. This presence of local levels of detection is directly affecting the (potential) size of the area of detectable change. Detectable change can only be assigned for points which have a precision estimate in both point clouds. This is highly dependent on the point distribution in each sparse point cloud as the point precision estimates are computed on the sparse point cloud and later interpolated to the dense point cloud. The sparse point clouds of the aerial imagery seem well distributed over the sinkhole, allowing a good interpolation of the values on the dense point cloud, resulting in a good spatial coverage. However, the terrestrial sparse point cloud points are less well distributed, which resulted in dense cloud points missing precision estimates, given our applied interpolation method, only allowing limited areas of extrapolation. Generally, our resulting precision map patterns may be interpreted as variations in precision estimates originating from the randomness in the photogrammetry rather than from the ground control points [38]. The 2019 aerial point cloud is an exception to this, as the precision map shows a distinct gradient which may be attributed to weak GCPs [38]. However, the same GCPs were used for the 2019 terrestrial point cloud which shows a different pattern in the precision maps. The very low point precision estimates of the 2017 aerial point cloud may be attributed to a very strong image network geometry with pictures taken from different distances from well around and within the sinkhole, as recommended by Micheletti et al. [3], and to a good ground sampling distance (Figure A1). In general, the aerial point clouds show a lower RMSE of the ground control points, which is related to the closer ground sampling distance of the aerial imagery, a relationship also found by Smith and Vericat [21] and Eltner et al. [4].

### 4.2. Challenges in Multitemporal/Multisensor Comparison

In multitemporal comparison changes, such as different locations of GNSS surveyed points, illumination differences and vegetation growth pose challenges, which we discuss in the following.

Due to the site location, potential erosion on the sinkhole edge, and limited possibilities of installing semipermanent georeference targets, the location of the base station and the ground control points (GCPs; targets) changed slightly in between the years. Additionally, it was not possible to place independent control points inside the sinkhole itself due to potentially hazardous conditions and missing equipment such as a total station.

Additionally, we found the manual setting of the markers much easier in the aerial imagery than the terrestrial imagery (due to better viewing angles and resolution). Potential uncertainties arising from these circumstances might be eliminated by applying the Time-SIFT method [58]. It could be applied to reduce the work and potential errors of manually setting the markers for each year, potentially reducing errors in coregistration and to have more CPs available to analyze the quality of the 3D reconstruction. This Time-SIFT, or also a refined coalignment approach [52], could be of particular interest for practitioners to reduce the work during each survey while still accomplishing good-quality point clouds.

The success of the feature detection within images necessary for the 3D reconstruction is highly dependent on the success of identifying matching unique features in the image set [59]. This is reportedly influenced by anything that makes the unique features appear differently in between images: the image quality and resolution, number of images, changes in illumination (e.g., arising from changes in the sun position and shadows or filtering by clouds resulting in potential over-or underexposure of (parts of) images), and changes in the observed object (e.g., wind shifting vegetation [5,20]). While the image quality and number can be controlled easily in between surveys, the changes in the illumination and within the observed object are challenging in multitemporal/multisensor studies in natural settings. Ideally, the images during each year would be collected under very similar illumination conditions. However, this is not always possible to obtain when scheduling field work. Since our study objective was comparing aerial and terrestrial surveys for change detection, the most important part of the experiment design was to acquire imagery on the same date, with no erosion and deposition occurring between the surveys, which we did achieve. However, in our study, challenges were posed by changing illumination (sun and shadows causing over- und underexposure of parts of the images) during the survey and in between the survey years and by wind shifting vegetation and vegetation growth. We assume that shadows occurring in the sinkhole (in 2017 and 2019) resulted in over-or underexposure in the images, which affected the 3D reconstruction and potentially the precision of the resulting point cloud (2017 Terr [27]). Additionally, different illumination can influence the identification of the targets in the images, and, therefore, the quality of the GCP marking, as we noticed with overexposure due to sunlight in the 2017 imagery.

Due to windy conditions, the moving vegetation (or parts of it) was not reconstructed properly, which, combined with the gradual filtering, resulted in holes of the 3D point cloud mainly at the bottom of the sinkhole. Additionally, we observed a good growth of the vegetation within parts of the sinkhole over the years. Obstruction by vegetation is challenging for optical sensors, as the wavelengths are not able to penetrate the foliage, resulting in local loss of surface information in vegetated areas and highly irregular surfaces with holes. While we are aware that growing vegetation has a large influence on the change detection of the slopes and the sinkhole floor, we still decided to keep the sinkhole undisturbed from artificial processes such as our survey. As the vegetation growth was not of interest for our study we looked for ways of eliminating the vegetation from the point clouds. Our approach of reducing the vegetation within the point clouds by using the gradual filtering tool was only partly successful: vegetation growing higher from the ground (small trees or higher plants with stems) was reduced to mainly the stems; vegetation growing close to the ground could not be removed. Here, other approaches, such as the usage of the local variance [12], automatic classification of the point cloud (vegetation and no vegetation [60]), adapting a cloth simulation filter [61], or more time-consuming manual filtering of the vegetation from the point clouds, might be promising. However, the different appearance (color) of the vegetation in all three years and the complex terrain are expected to pose challenges for automated filtering processes.

## 5. Conclusions

Our comparison of terrestrial and aerial SFM surveying of the sinkhole showed that using imagery from aerial surveying results in better-quality point clouds. The point clouds resulting from the aerial imagery show the best spatial coverage of the sinkhole, the lowest

point precision estimate range, and the best spatial coverage with point precision estimates. However, given an ideal surveying and image-taking method, the terrestrial imagery still results in good quality and interpretable point clouds fit for change detection, as shown with the 2019 survey. This is confirmed by the comparisons of the observed patterns of change in between the years which were found to be comparable between using terrestrial or aerial imagery for computing the point clouds. Therefore, it depends on the intended scope, scale of measured changes, and aimed local level of detection as to which survey method may be chosen in similar surveys.

Furthermore, although one might be under the impression that it is possible to obtain a good point cloud from any set of images, we need to conclude that the spacing between image locations and degree of angular changes between images are decisive to obtain a reliable point cloud, fit for multitemporal comparisons.

The point precision estimates and derived local levels of detection were valuable sources for correctly interpreting the measured distances within the sinkhole. However, a clear limitation of this approach is the dependence on the sparse point cloud and its interpolation to the dense point cloud. Consequently, for points with no precision estimates, no detectable change can be assigned.

**Author Contributions:** Conceptualization, H.P. and J.G.; data curation, M.Z. and J.G.; formal analysis, H.P. and M.Z.; funding acquisition, H.P.; methodology, H.P. and J.G.; resources, H.P.; supervision, H.P. and J.G.; visualization, H.P.; writing: original draft, H.P. and M.Z.; writing: review and editing, H.P., M.Z., P.F. and J.G. All authors have read and agreed to the published version of the manuscript.

**Funding:** This research was partly funded by the Hanna Bremer Award for young female physical geographers of the Hanna Bremer Foundation, awarded by Helene Petschko in 2016.

**Data Availability Statement:** The data presented in this study are openly available in Zenodo at https://doi.org/10.5281/zenodo.6521706.

**Acknowledgments:** The authors thank Clemens Paulmann, Ronja Prietzsch, Theo Stahnke, Luka Borgelig, and Christian Pfeifer for their help during fieldwork. We are very grateful for the IT support from Andreas Schäf.

**Conflicts of Interest:** The authors declare no conflict of interest. The funders had no role in the design of the study; in the collection, analyses, or interpretation of data; in the writing of the manuscript, or in the decision to publish the results.

## Appendix A

*Appendix A.1. Camera Positions and Image Overlap*

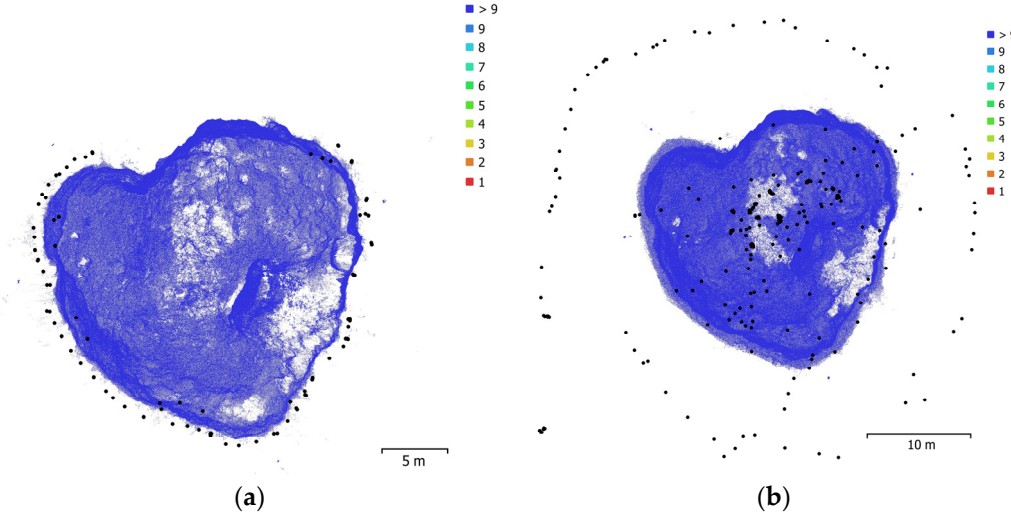

(a)                                        (b)

**Figure A1.** *Cont.*

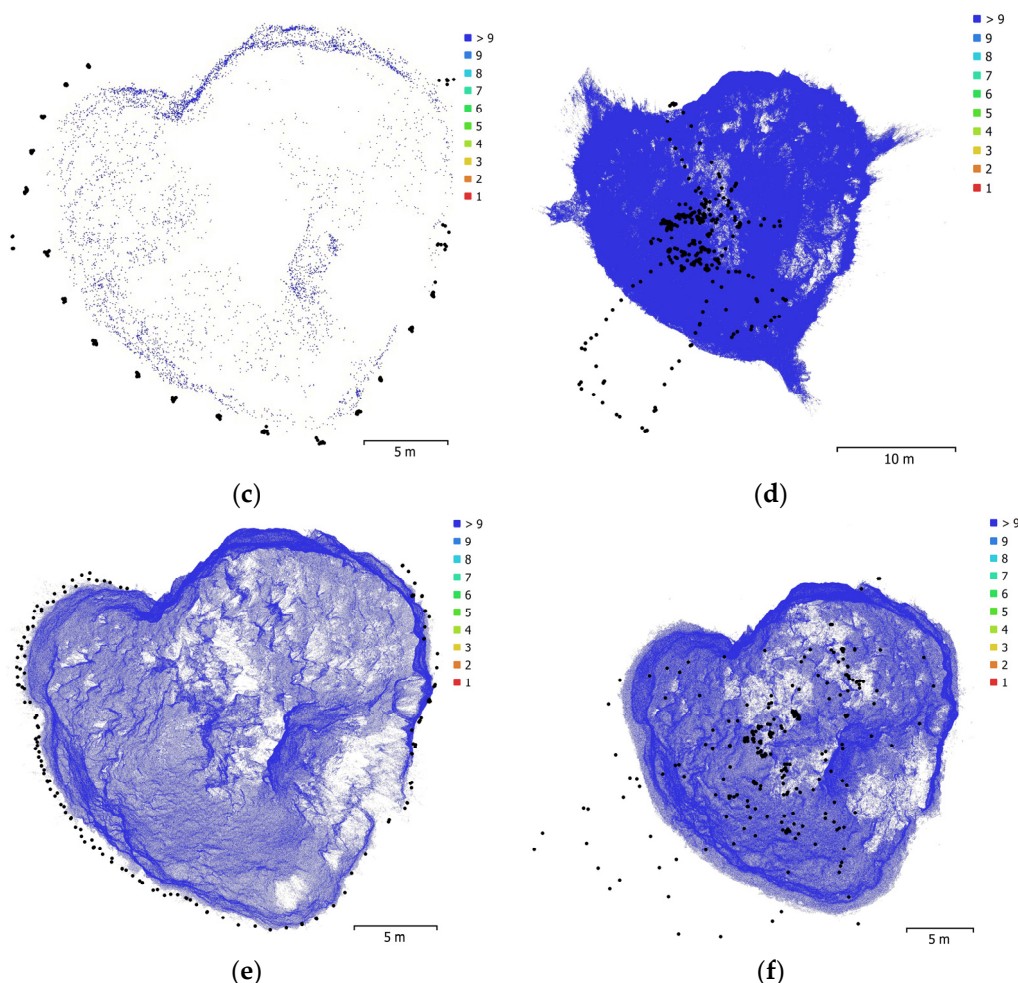

**Figure A1.** (**a**) 2017 point cloud from terrestrial imagery, (**b**) 2017 point cloud from aerial imagery, (**c**) 2018 point cloud from terrestrial imagery, (**d**) 2018 point cloud from aerial imagery, (**e**) 2019 point cloud from terrestrial imagery and (**f**) 2019 point cloud from aerial imagery. (**a**–**e**): Camera positions and image overlap as resulting from the Agisoft Photoscan 3D reconstruction and project report. The black dots give the camera position. The colors describe the number of overlapping images.

*Appendix A.2. Point Cloud Coverage of the Sinkhole*

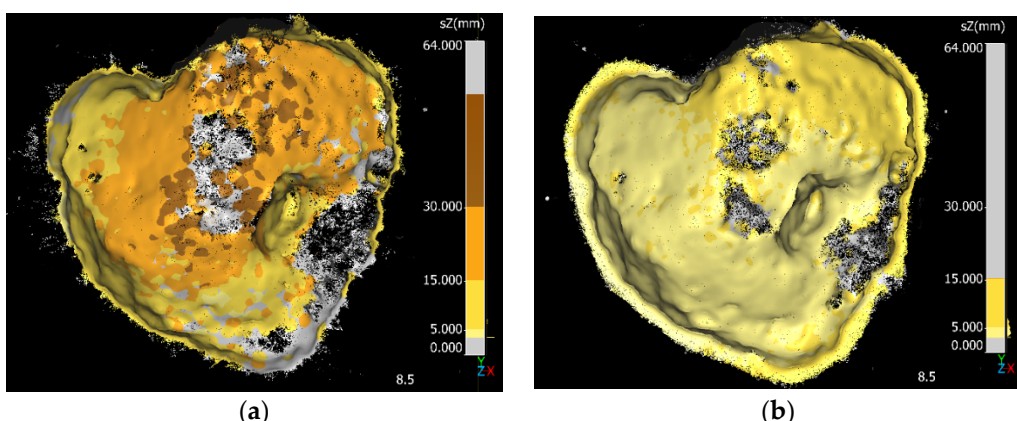

**Figure A2.** *Cont.*

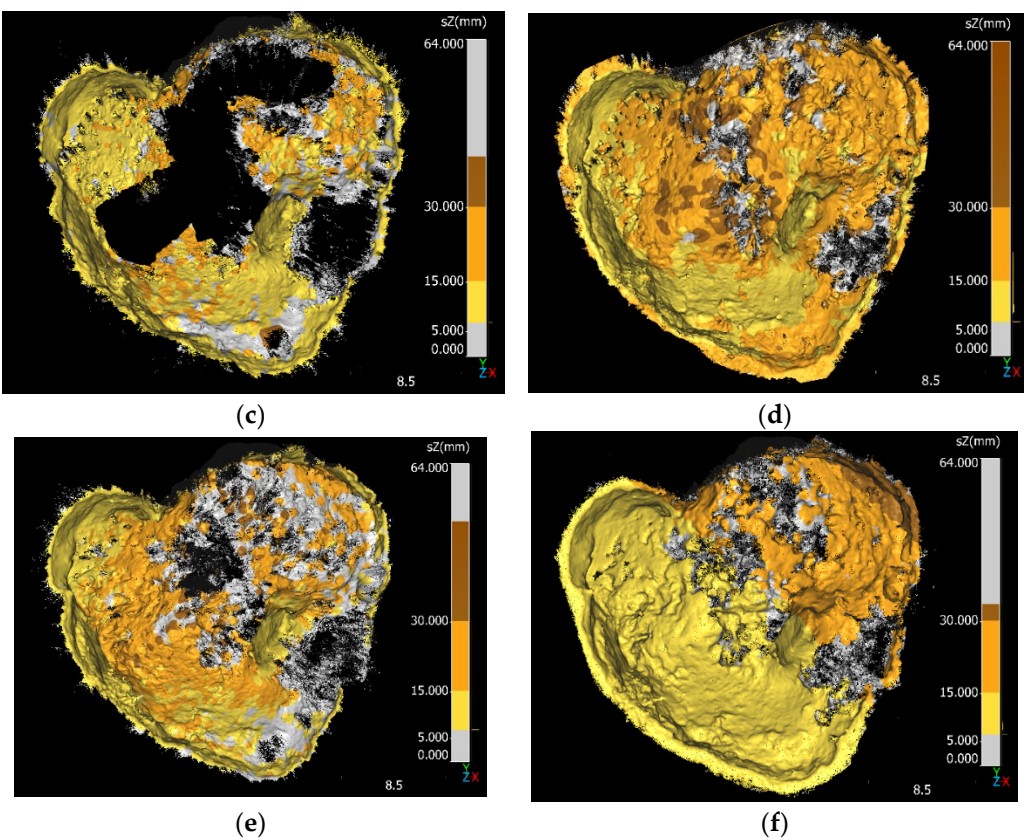

**Figure A2.** (**a**) 2017 point cloud from terrestrial imagery, (**b**) 2017 point cloud from aerial imagery, (**c**) 2018 point cloud from terrestrial imagery, (**d**) 2018 point cloud from aerial imagery, (**e**) 2019 point cloud from terrestrial imagery and (**f**) 2019 point cloud from aerial imagery. (**a**–**e**): Input PNG pictures of the point clouds; view from the top. Note the considerable amount of black pixel around the sinkhole. As this is assumed to be roughly the same size for all point clouds, assessing the percentage of black pixel is assumed to be a good measure of the different sizes of holes in the point clouds. As scalar field we decided to visualize the vertical point precision estimates as in Figure 2 as well.

*Appendix A.3. Local Levels of Detection of Comparison Pairs*

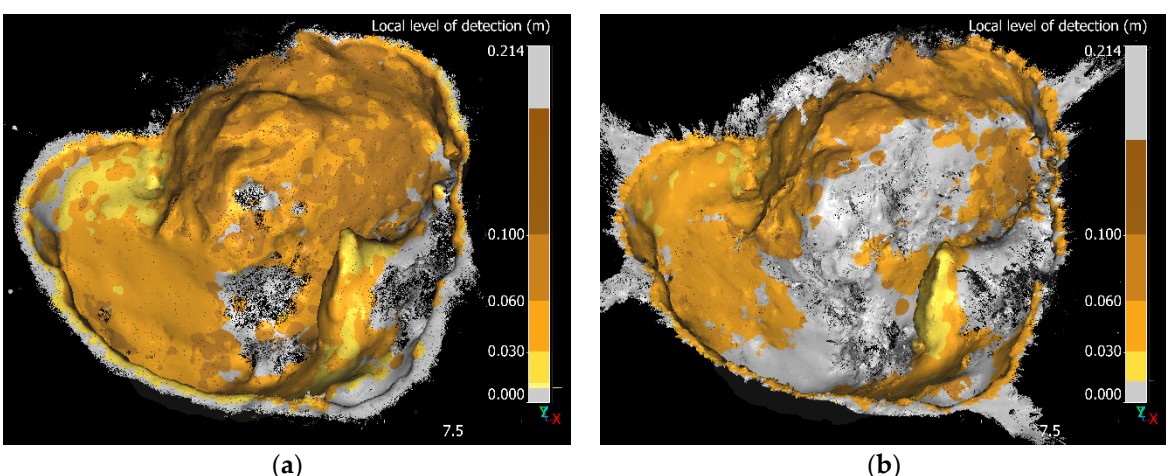

**Figure A3.** *Cont.*

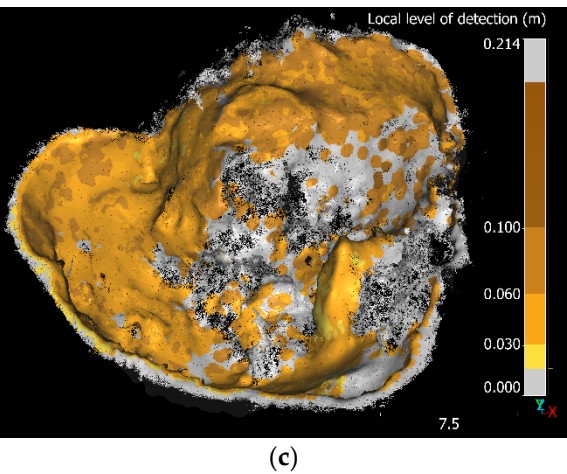

(**c**)

**Figure A3.** (**a**) 2017 Terr UAV, (**b**) 2018 Terr UAV and (**c**) 2019 Terr UAV. (**a**–**c**): Local level of detection for the comparison of point clouds resulting from using terrestrial or aerial imagery. Gray colors in the point clouds reflect areas with no information point precision estimates available in any of the point clouds of the comparison pair. In the color scale the gray areas show that there are no values available, therefore they are indicating the range of the local levels of detection of the respective comparison pair.

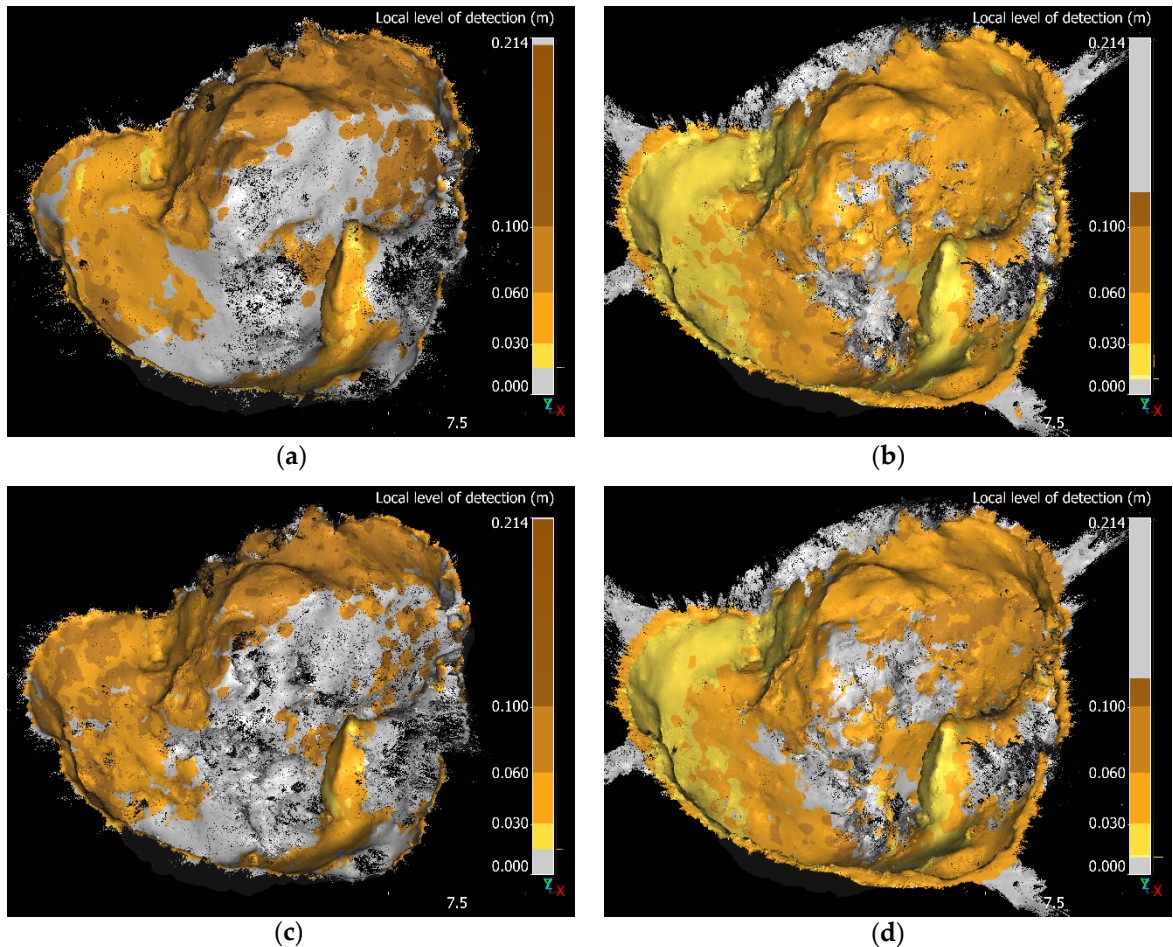

(**a**)                                   (**b**)

(**c**)                                   (**d**)

**Figure A4.** *Cont.*

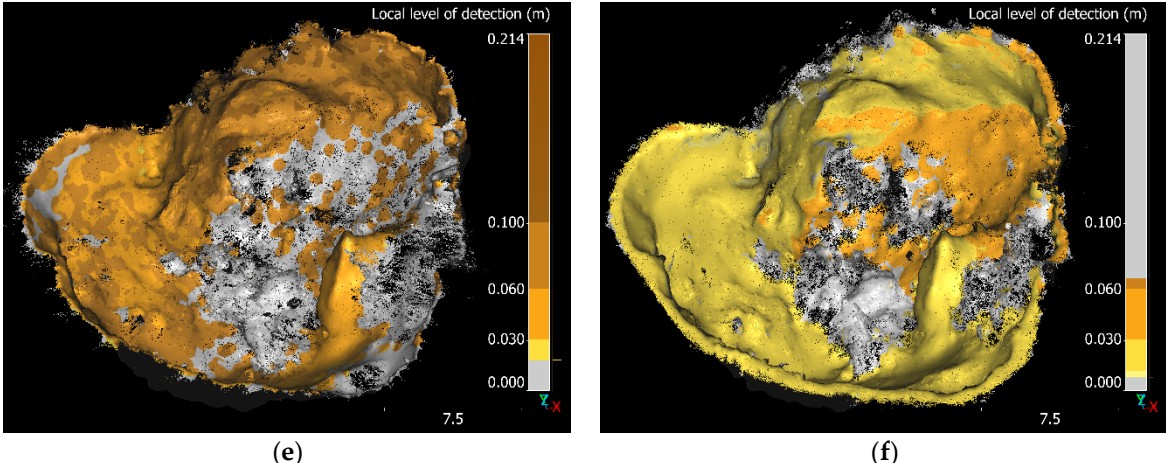

(**e**)  (**f**)

**Figure A4.** (**a**) 201718 Terr, (**b**) 201718 UAV, (**c**) 201819 Terr, (**d**) 201819 UAV, (**e**) 201719 Terr and (**f**) 201719 UAV. (**a**–**f**): Local level of detection for the multitemporal comparison of point clouds resulting from using terrestrial and aerial imagery in the 3D reconstruction. Gray colors in the point clouds reflect areas with no information point precision estimates available in any of the point clouds of the comparison pair. In the color scale the gray areas show that there are no values available, therefore they are indicating the range of the local levels of detection of the respective comparison pair.

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
