# Peer review of "Terrestrial and Airborne Structure from Motion Photogrammetry Applied for Change Detection within a Sinkhole in Thuringia, Germany"

_remotesensing, doi:10.3390/rs14133058_

Round 1

Reviewer 1 Report

The presented paper provides an interesting view on the comparison of terrestrial and UAV SfM photogrammetry applied for change detection in geomorphological applications.

The scientific novelty of such a comparison might be questionable since there are many papers describing this topic by various approaches. However, I have to highlight the overall concept of the paper and the detail and quality of the point clouds comparison and deformation analysis. In my opinion, many professionals, as well as scientists from various application fields such as geomatics, photogrammetry, geomorphology, earth sciences, etc., can find useful information here.

I have only minor comments and questions on the paper:

1. I would stick to the term "image" instead of "photo", or "photograph". Although not essential, it is often used in preference to photograph.

2. Figure 1 is not so well readable, mainly the sinkhole location. Please provide a more illustrative image for the location; for instance, within Germany; within Europe, etc.

3. Table 1 - The illumination conditions and ground/weather conditions are indeed quite different between the three epochs. Did you consider under/overexposure (some of the UAV images, mainly in 2018), or sun/shadow changes (mainly in 2019)? Did you consider doing the imaging under as similar conditions as possible (for instance, to also avoid snow patches)? Did these different conditions affect the results somehow?

4. Lines 195-200: Why did you not use coded targets that can be generated and subsequently automatically detected with high accuracy in Agisoft software?

5. Regarding GCPs, is it enough to use such positions of GCPs? For terrestrial images, almost all points are situated only below the top edge of images; with a very low angle of the viewing direction. This can result in the lower accuracy values (Tab. 3), and even cause the "doming" effect in the final point cloud - as you are stating in the Discussion. However, with better GCPs positioning, you can get a clearer view of this issue.

Maybe you could place some targets on a solid board, hang over the edge to a suitable depth and measure its coordinates by the total station?

6. Lines 238-239: How exactly did you exclude not-good images? Did you use the function provided in Agisoft to "Estimate Image Quality"?

7. Lines 252-254: Why did you use such a high threshold values for projection accuracy in gradual filtering (10 and 15)? In my experience, for high accuracy projects, one should go for as low as 3 for consumer graded UAV cameras, or even lower.

Reviewer 2 Report

This work is devoted to the comparison of two methods of surveying to determine surface changes occurring in sinkholes. In order to evaluate the possibility of using ground-based photography, the authors used multiscale model to model cloud comparison using precision maps plugin (M3C2-PM) in CloudCompare, that allowed to determine the differences between the point clouds arising from the different sensors and data collection methods per year.

The article is of some scientific interest in the application of the mentioned cloud analysis algorithm, but I have some remarks and comments on the surveying and photogrammetric processing.

1.     The article does not specify GSD for each type of survey. As is known, the accuracy of x,y,z coordinate determination by photogrammetric method depends on the survey parameters: focal length f, GSD, image baseline b and distance to the object, determined by the ratio (f/b)*GSD. Thus, the assumption that the point clouds obtained by aerial survey and ground survey should give the same results is incorrect (line 438).

2.     The text (line 235) mentions that control point measurements were made in stereo mode in anaglyph mode. It is not explained for what purpose this was done. Marked points are perfectly detected by automatic identification algorithms based on overlapping images

3.     Usually during photogrammetric processing the estimation of alignment accuracy is performed by control points. Coordinates of these points are not used for external model orientation, but they are used to estimate the real georeferencing accuracy.

4.     Filtering of tie points can lead not to improvement of phototriangulation, but to the opposite result. The reason is uneven distribution of the remaining points on the image. This needs to be controlled.

5.     This method can detect changes in topography that are the same size or larger than the GSD.
